# White matter tracts adjacent to the human cingulate sulcus visual area (CSv)

**Maiko Uesaki** [1,2,3]*, **Michele Furlan**[4], **Andrew T. Smith**[5], **Hiromasa Takemura** [1,6,7]*

**1** Center for Information and Neural Networks (CiNet), Advanced ICT Research Institute, National Institute of Information and Communications Technology (NICT), Suita, Osaka, Japan, **2** Graduate School of Frontier Biosciences, Osaka University, Suita, Osaka, Japan, **3** Open Innovation & Collaboration Research Organization, Ritsumeikan University, Ibaraki, Osaka, Japan, **4** Scuola Internazionale Superiore di Studi Avanzati (SISSA), Trieste, Trieste, Italy, **5** Department of Psychology, Royal Holloway, University of London, Egham, Surrey, United Kingdom, **6** Division of Sensory and Cognitive Brain Mapping, Department of System Neuroscience, National Institute for Physiological Sciences, Okazaki, Aichi, Japan, **7** Graduate Institute for Advanced Studies, SOKENDAI, Hayama, Kanagawa, Japan

* uesaki@nict.go.jp (MU); htakemur@nips.ac.jp (HT)

**Data Availability Statement:** Authors have made materials and code required to replicate graphs and statistical analyses included in this study available to the public (https://osf.io/sfamu/). The MRI data analysed in this study were originally acquired by

## Abstract

Human cingulate sulcus visual area (CSv) was first identified as an area that responds selectively to visual stimulation indicative of self-motion. It was later shown that the area is also sensitive to vestibular stimulation as well as to bodily motion compatible with locomotion. Understanding the anatomical connections of CSv will shed light on how CSv interacts with other parts of the brain to perform information processing related to self-motion and navigation. A previous neuroimaging study (Smith et al. 2018, Cerebral Cortex, 28, 3685–3596) used diffusion-weighted magnetic resonance imaging (dMRI) to examine the structural connectivity of CSv, and demonstrated connections between CSv and the motor and sensorimotor areas in the anterior and posterior cingulate sulcus. The present study aimed to complement this work by investigating the relationship between CSv and adjacent major white matter tracts, and to map CSv's structural connectivity onto known white matter tracts. By re-analysing the dataset from Smith et al. (2018), we identified bundles of fibres (i.e. streamlines) from the whole-brain tractography that terminate near CSv. We then assessed to which white matter tracts those streamlines may belong based on previously established anatomical prescriptions. We found that a significant number of CSv streamlines can be categorised as part of the dorsalmost branch of the superior longitudinal fasciculus (SLF I) and the cingulum. Given current thinking about the functions of these white matter tracts, our results support the proposition that CSv provides an interface between sensory and motor systems in the context of self-motion.

## Introduction

Accurate perception of self-motion lays the basis for much of our interaction with and navigation within the environment. It allows us, for example, to avoid obstacles, to control our posture, and to maintain the intended trajectory during locomotion. For primates, including

Smith et al. (2018), for which the informed consent obtained did not include authority for public data release. For this reason, we are unable to make the raw MRI data available to the public. Any request for the raw MRI data should be directed to the corresponding author of the original paper (A.T.S.: a.t.smith@rhul.ac.uk) in the first instance, and if necessary to the secretary of the Royal Holloway Ethics Committee (ethics@rhul.ac.uk).

**Funding:** This work was supported by Japan Society for the Promotion of Science (JSPS; https://www.jsps.go.jp/) KAKENHI (JP17H04684 and JP21H03789, H.T.), Grant-in-Aid for JSPS Fellows (JP13J05795, M.U.), and Research Grant for Nanyang Technological University Presidential Postdoctoral Fellows (https://www.ntu.edu.sg/research/research-careers/presidential-postdoctoral-fellowship-(ppf)). The funders had no role in study design, data collection and analysis, decision to publish, or preparation of the manuscript.

**Competing interests:** The authors have declared that no competing interests exist.

humans, one of the most important sensory cues that enables the perception of self-motion is optic flow. Optic flow refers to distinct patterns of motion projected onto the retina as we move through an environment [1], and provides information that can be used to compute the direction of heading [2, 3] as well as time to contact [4] and distance travelled [5]. Understanding the neuronal circuitry underlying optic-flow processing is therefore essential in order to elucidate the mechanisms that enable us to monitor and adjust our locomotory movements according to the changes in the external environment.

Electrophysiological studies on macaque monkeys first revealed that neurons in the dorsomedial part of area MST (MSTd) respond selectively to optic flow [6–8]. Evidence has since accumulated, based on functional magnetic resonance imaging (fMRI) of both macaque and human brains, that the neuronal representation of optic flow is not restricted to MSTd, but rather distributed amongst several areas in the occipital, parietal and cingulate cortices [9–12]. Interestingly, some of those cortical areas have also been associated with vestibular- [13, 14] and motor-related [15, 16] activity, suggesting that visual, vestibular and motor signals may combine in these cortical regions to support self-motion and its perception.

The cingulate sulcus visual area (CSv; [9, 10] see [17] for a review) is one of the cortical areas that respond selectively to optic flow compatible with self-motion. CSv is located bilaterally in the posterior part of the mid-cingulate sulcus. Since it was first described in Wall and Smith [9], numerous studies have demonstrated CSv's strong specificity to visual self-motion [9, 18–20], and its involvement in monitoring self-motion [21, 22]. Evidence that further supports CSv's role in encoding visual cues to self-motion and guiding locomotion comes from findings that CSv receives vestibular input [13, 23, 24], and that CSv is activated during lower-limb, but not upper-limb movements [15, 16]. However, a question remains as to how CSv interacts with other cortical areas via structural connections within the white matter.

Recent advances in diffusion-weighted MRI (dMRI) methods have provided opportunities to non-invasively study the white matter connections in living human brains, and compare them with functionally defined areas measured by fMRI [25–27]. Diffusion MRI measures anisotropy of water diffusion in brain tissues. Because water molecules preferentially move parallel to fibre bundles in the white matter, dMRI data can be used to estimate the fibre orientation within each white matter voxel. By applying tractography, which traces fibre orientations across white matter voxels, the three-dimensional trajectory of estimated fibre bundles (i.e. streamlines; [28, 29]) can be reconstructed. Smith and colleagues [30] used this approach to investigate the structural connectivity patterns of CSv. They defined CSv with a functional localiser and used it as a seed region for connectivity analyses based on dMRI measurements of the same human brains. They examined the cortical distribution of streamline endpoints and found that CSv is connected ipsilaterally with the motor and sensorimotor areas in the anterior and posterior cingulate sulci respectively, and contralaterally with the paracentral gyrus and sulcus. A similar approach was applied in macaques by De Castro et al. [11], which demonstrated that the putative macaque CSv is connected with the cingulate sulcus, much like in humans. However, because those studies focussed on the cortical endpoints of streamlines, their results did not reveal the trajectories of CSv streamlines within the white matter, thus the white matter structure involved in signal transmission to and from CSv remains to be explained.

Anatomical studies have shown that the white matter consists of a number of major bundles of axons (i.e. tracts), such as the superior longitudinal fasciculus (SLF) and cingulum (see [31–33] for reviews). Those tracts pass through specific portions of the white matter, and it has been found that lesions to specific white matter tracts lead to impairment of specific brain functions [34]. Recent studies have also demonstrated that different subdivisions of the same white matter tract (e.g. three branches of SLF; SLF I, II, and III) show different degrees of

lateralisation as well as age dependency, and correlation with cognitive functions and behaviour [35–37]. Therefore, understanding the relationship between the white matter tracts and CSv connectivity will provide essential insights into the position of CSv in the context of established properties of the white matter tracts and how their lesions might impact brain functions.

This study re-analysed the data from Smith et al. [30], which included both fMRI data allowing for CSv localisation and dMRI data for tractography. In contrast to Smith et al. [30], we focussed on identification and characterisation of the white matter tracts around the functionally defined CSv. We employed dMRI-based tractography and waypoint region of interest (ROI) approach [38–40], and analysed the relationship between CSv and major white matter tracts by categorising the streamlines terminating around CSv based on the white matter tracts to which they belong. We also applied filtering to generated streamlines, which resulted in more conservative estimates of structural connectivity supported by the dMRI data [41–44].

## Materials and methods

We analysed fMRI and dMRI data acquired from healthy human subjects in previous work (see [30] for details). All MRI data were collected with a 3T Siemens TIM Trio MAGNETOM MRI scanner (Siemens, Erlangen, Germany) equipped with a 32-channel head coil at Royal Holloway, University of London.

### 1. Subjects

Twelve healthy volunteers (five males and seven females; median age 23.5 years; S1-S12) participated in the study in accordance with the ethical standards stated in the Declaration of Helsinki, and approval from the Royal Holloway Research Ethics Committee.

### 2. Structural MRI data acquisition and analysis

For each subject, a 3D T1-weighted structural MR image was acquired. The T1-weighted image was acquired with the modified driven equilibrium Fourier transform (MDEFT; [45]) sequence (160 sagittal slices, 1 mm isotropic voxels), and was used to perform tissue segmentation between the grey and white matter using FreeSurfer ([46]; https://surfer.nmr.mgh. harvard.edu/).

### 3. fMRI data acquisition and analysis

**1. Data acquisition.** Functional MRI data were acquired using a previously established localiser to identify CSv [9, 10]. The localiser consisted of two time-varying optic-flow stimuli. The first optic-flow stimulus cycled smoothly through spiral space to simulate back-and-forth rotational motion of the observer (i.e., compatible with self-motion). The second was a 3 x 3 array of similar spiral motions that was incompatible with self-motion. Visual stimuli were projected onto an in-bore rear-projection screen in the scanner, viewed via a monocular magnifying optical device placed over the subject's preferred eye. Each stimulus was presented for 3 s in an event-related design, with intertrial intervals varying between 2 and 10 s. Each of six fMRI scans consisted of 32 trials (16 per condition) presented in a pseudorandom order, and lasted approximately five minutes. Subjects maintained central fixation and were engaged in an attentional task at fixation throughout the scans.

Data were acquired with the generalised autocalibrating partially parallel acquisition (GRAPPA; [47]) sequence (36 slices, 3-mm isotropic voxels, time of repetition [TR]: 2500 ms,

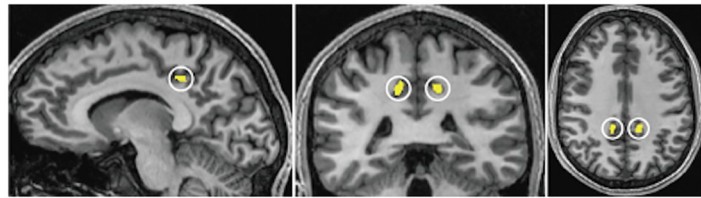

**Fig 1. Location of CSv (yellow patches highlighted by white circles) overlaid on the sagittal (left), coronal (middle), and axial (right) sections of one representative subject (S1).** See S1 Fig for other subjects.

time echo [TE]: 31 ms). Further technical details on fMRI data acquisition are provided in Smith et al. [30].

**2. Localisation of CSv.** We used CSv as identified in Smith et al. [30] by fMRI-based localiser data for subsequent analysis (see Fig 1 and S1 Fig). Functional MRI data were preprocessed and analysed with the general linear model (GLM) using BrainVoyager QX ([48]; version 2.3; Brain Innovation, Maastricht, The Netherlands), according to the methods previously described in Wall and Smith [30]. The region of interest, CSv, was defined as contiguous voxels, of which response was significantly greater to a single optic-flow patch than to an array of optic-flow patches, within the cingulate sulcus in each hemisphere. As reported in Smith et al. [30], the clusters of voxels belonging to CSv combined across all subjects had centres of gravity of [9 -24 44] and [-10 -26 41] in MNI152 space in the left and right hemispheres, respectively.

## 4. dMRI data acquisition and analysis

**1. Data acquisition.** Diffusion-weighted MRI data were acquired in each subject by a spin-echo sequence with echo-planar readout (65 axial slices, 2-mm isotropic voxels, TR: 9300 ms, TE: 94 ms, field of view: 192 x 192 mm$^2$). The diffusion-weighted data were sampled in 64 directions using a b-value of 1000 s/mm$^2$. Three sets of diffusion-weighted data were acquired for all but one subject. For technical reasons, only two sets were acquired in one subject. The results for this subject were not anomalous. In addition, six low b-value ($b = 0$) images were obtained, in pairs interleaved with the three sets of diffusion-weighted scans. Further technical details on dMRI data acquisition are provided in Smith et al. [30].

**2. Data preprocessing.** Diffusion-weighted MRI data were preprocessed using mrDiffusion, implemented in Matlab as part of vistasoft software distribution (https://github.com/vistalab/vistasoft). Diffusion-weighted images were corrected for eddy currents and head motion by a 14-parameter constrained non-linear coregistration [49]. The gradient direction in each volume was corrected using the rotation parameters from the motion-correction and eddy-current correction processes. The diffusion tensor model was fitted to dMRI data to generate a colour-coded principal diffusion direction (PDD) map to visualise diffusion direction in each voxel. In addition, fibre orientation distribution in each voxel was estimated by constrained spherical deconvolution (CSD; [50]; $L_{max} = 8$) using MRtrix3 ([51, 52]; http://www.mrtrix.org/).

**3. Fibre tracking and evaluation.** We used the ensemble tractography method [44]. This minimises the known dependency of tractography on the choice of parameters [53–56] by generating streamlines using multiple tractography parameters. Amongst the streamlines generated, those that did not align with diffusion signals along the trajectories were then culled using linear fascicle evaluation (LiFE; [41, 57]; http://francopestilli.github.io/life/), which is

one of the methods developed to filter and remove spurious streamlines. Our approach aimed to achieve a more conservative selection of streamlines, while reducing the dependency of tractography on arbitrary choices of parameters.

Specifically, we generated eight million candidate streamlines using CSD-based probabilistic tractography implemented in MRtrix3 (iFOD2; [58]) using four angle thresholds (5.7, 11.5, 23.1, and 47.2 deg; two million candidate streamlines per angle threshold). We set the minimum streamline length at 4 mm and the maximum at 250 mm. The default parameters of MRTrix3 CSD-based probabilistic tractography were used except for the maximum streamline length (250 mm). The seed voxels for tracking were randomly chosen from the grey-white matter interface region [59]. Finally, LiFE; [41, 57]) was applied to filter streamlines. This process removed streamlines that did not contribute to predicting diffusion signals and therefore yielded the optimal set of streamlines (optimised streamlines). In subsequent analyses, we considered only the optimised streamlines. Technical details of the ensemble tractography method are described in previous works [26, 44, 60].

## 5. Identification of white matter tracts adjacent to CSv

In order to identify white matter tracts near CSv, we first selected a subset of streamlines near CSv from whole-brain streamlines optimised with LiFE. We selected streamlines that have an endpoint falling within a threshold distance (3 mm) from CSv voxels identified using the fMRI localiser. Fig 4 shows CSv streamlines in a representative subject.

We then examined how CSv streamlines run into major white matter tracts. We selected seven candidate major white matter tracts (SLF I, II, III, the cingulum, the callosal fibres, the arcuate fasciculus, and the corticospinal tract), whose proximity to CSv suggests that they could be associated with CSv, by comparing the coordinates of CSv and the positions of major white matter tracts described in the established atlases of the white matter [31, 33]. We used the waypoint ROI approach to segment subsets of CSv streamlines belonging to the major white matter tracts (Fig 2), following the criteria as in previous work [35, 37, 40, 61]. Five of these tracts were manually segmented, due to difficulty in delineating neighbouring tracts using automated methods. Manual segmentation of the white matter tracts also ensured consistency with previously established protocols based on anatomical information [35, 37, 61]. The detailed procedure is described below.

**1. Superior Longitudinal Fasciculus (SLF) I, II, and III.**   We identified CSv streamlines turning into SLF I, II, and III using the coronal waypoint ROIs. The waypoint ROIs were manually drawn for each subject on the coronal slice with the anterior commissure (AC), following the definitions used in previous studies [35, 37, 61]. Specifically, we manually drew the waypoint ROIs, each of which covered an area in the white matter in the superior frontal, middle frontal, and precentral gyri for SLF I, II, and III, respectively, in each hemisphere (Fig 2, cyan for SLF I; blue for SLF II; purple for SLF III). We used the following sulcus landmarks to draw the waypoint ROIs: the cingulate and the superior frontal sulci defined the borders of the waypoint ROI for SLF I, the inferior precentral sulcus was used as a landmark for the border between waypoint ROIs for SLF II and III, and the lateral fissure defined the inferior border of the waypoint ROI for SLF III. We defined CSv streamlines passing through those ROIs as streamlines belonging to SLF I, II, and III, respectively.

**2. Cingulum.**   Similarly, we identified CSv streamlines turning into the cingulum using a coronal waypoint ROI (Fig 2). The waypoint ROI for the cingulum was manually defined in the same coronal slice as were ROIs for SLF. We defined the coronal ROI for the cingulum in the area of the white matter adjacent to the cingulate gyrus and ventral to the cingulate sulcus, as in previous studies [62, 63]. We used the orientation of the cingulate sulcus in the coronal

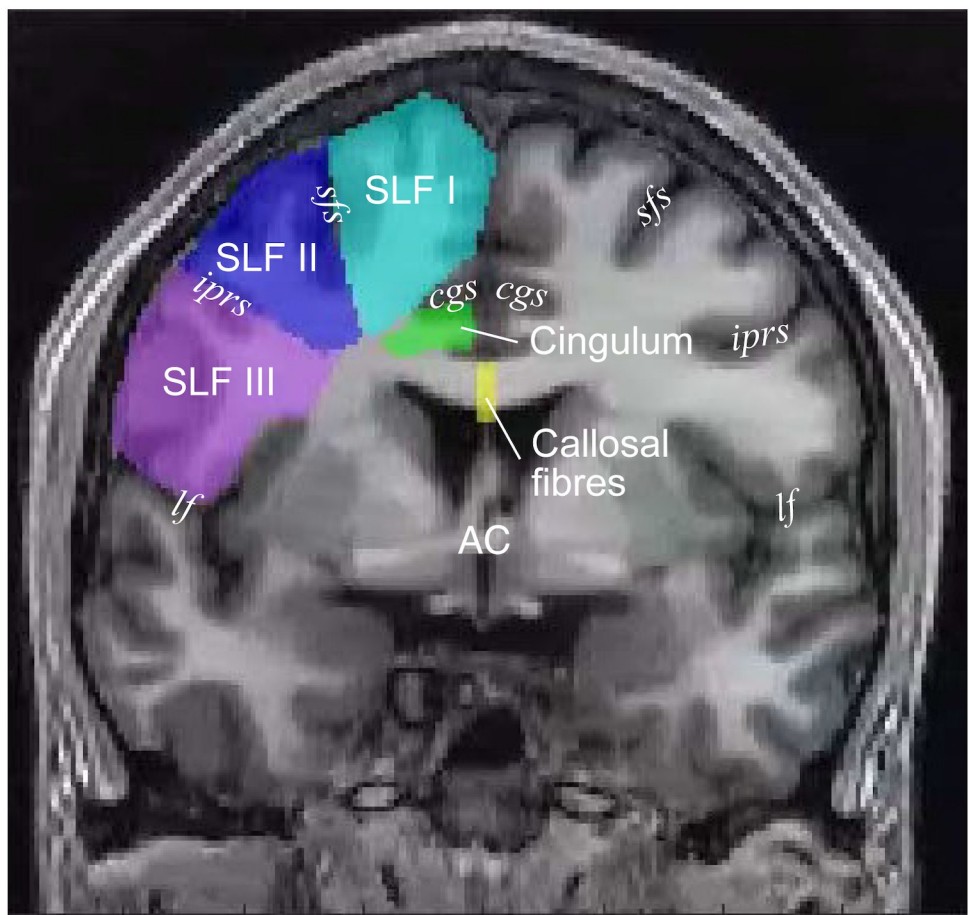

**Fig 2. Manually delineated waypoint ROIs used to identify five of the major white matter tracts analysed in this study, superimposed on a coronal section of the T1-weighted image of one representative subject (S1).** The coloured areas depict ROIs for the five tracts (cyan, SLF I; blue, SLF II; purple, SLF III; green, cingulum; yellow, callosal fibres). AC: anterior commissure, cgs: cingulate sulcus, sfs: superior frontal sulcus, iprs: inferior precentral sulcus, lf: lateral fissure, SLF: superior longitudinal fasciculus.

slice to define the border between ROIs for SLF I and the cingulum. CSv streamlines passing through this ROI were defined as part of the cingulum.

**3. Callosal fibres.** We identified CSv streamlines turning into the corpus callosum by defining a waypoint ROI for the callosal fibres in the sagittal slice (X = 0 in AC-PC coordinates). The waypoint ROI was manually drawn to cover the entire corpus callosum in the mid-sagittal slice (Fig 2). CSv streamlines passing through this ROI were defined as belonging to the callosal fibres.

**4. Arcuate fasciculus.** We identified CSv streamlines turning into the arcuate fasciculus (AF) using a waypoint ROI automatically generated by AFQ MATLAB toolbox [40]. The waypoint ROI covered an area of the white matter in the axial slice between the temporal and the fronto-parietal cortices [64]. CSv streamlines passing through this ROI were defined as belonging to AF.

**5. Corticospinal tract.** We identified CSv streamlines turning into the corticospinal tract (CST) using two waypoint ROIs automatically generated by AFQ MATLAB toolbox [40]. The inferior ROI was placed in the axial slice of the brainstem and covered the entire cerebral

peduncle. The superior ROI was placed in the axial slice and covered an area of the white matter composed predominantly of tracts with a superior-inferior trajectory, located inferior to the region where CST and callosal fibres cross each other. CSv streamlines passing through both of these ROIs were defined as belonging to the CST.

**6. Statistical analysis.** We evaluated the statistical significance of CSv streamlines belonging to each of the seven white matter tracts considered. We first calculated the proportion of CSv streamlines belonging to each tract. We then performed a one-sample t-test for each tract in each hemisphere. A null hypothesis of this statistical test is that the proportion of CSv streamlines belonging to each white matter tract is not different from zero. We defined a statistical significance ($\alpha$) as $P = 0.007$, which is equivalent to $P = 0.05$ after Bonferroni correction for seven tracts tested for each hemisphere.

## Results

This study aimed to identify and characterise CSv streamlines in relation to the white matter tracts located near the functionally defined CSv, by re-analysing fMRI and dMRI data originally acquired by Smith et al. [30]. CSv was localised in each subject by contrasting blood-oxygen level-dependent (BOLD) responses to optic flow compatible with self-motion against those to an array of optic-flow patches incompatible with self-motion [9]. CSv was identified in the posterior part of the mid-cingulate sulcus (Fig 1), in both hemispheres of all subjects (see [30] for details).

Functionally localised CSv was overlaid on the PDD map of dMRI data, to visualise the spatial relationship between CSv and major white matter tracts. PDD maps are widely used to identify the positions of white matter tracts without performing tractography [65–68]. Fig 3 shows CSv superimposed on the PDD map on a representative coronal slice of four subjects (S1-S4). Visual inspection of the PDD maps revealed that CSv is located near the cingulum, which can be seen as white matter voxels inferior to the cingulate sulcus with the anterior-posterior diffusion direction (green). It is, however, difficult to identify clear borders amongst other white matter tracts, such as SLF I, II, and III solely based on the PDD maps. In order to examine the structural connectivity of CSv in relation to the known white matter tracts with more specificity, we subsequently performed tractography on dMRI data.

Tractography was performed on dMRI data to identify streamlines that have endpoints near CSv. We note that we took a conservative approach and analysed only the optimised streamlines (see S2 Fig for a comparison between CSv streamlines with and without application of LiFE). As shown in Fig 4, CSv streamlines include long-range streamlines towards the frontal cortex, shorter-range streamlines towards the dorsal, lateral, and posterior parts of the parietal cortex as well as towards the temporal cortex, and interhemispheric streamlines into the contralateral hemisphere. We then categorised the streamlines terminating around CSv, based on the waypoint ROIs through which they passed, to evaluate the relationship between

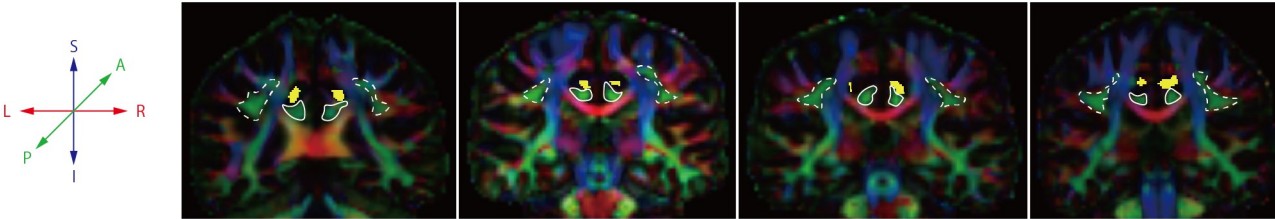

**Fig 3. Position of CSv (yellow) with respect to the PDD map of dMRI data on a representative coronal section of four subjects (S1-S4).** The colour scheme depicts the PDD in each voxel (blue, superior-inferior; green anterior-posterior; red, left-right). CSv is adjacent to the cingulum (highlighted by white solid lines), which can be seen as a group of voxels located inferior to the cingulate sulcus with the anterior-posterior diffusion direction.

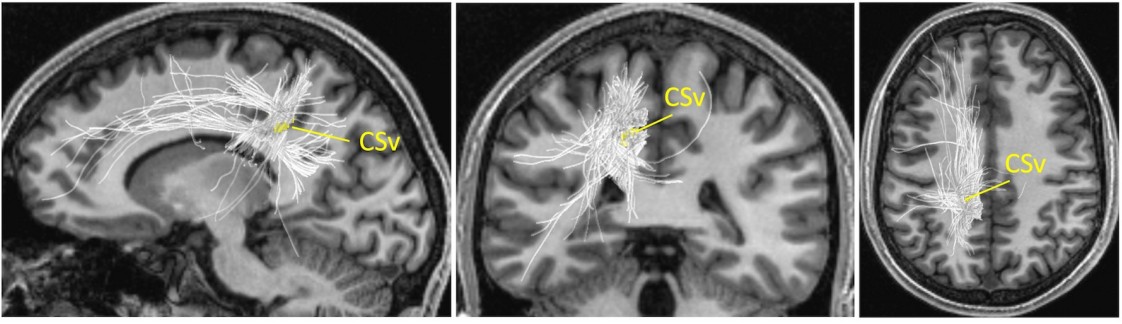

**Fig 4. CSv streamlines in the left hemisphere of one representative subject (S1).** Left CSv (yellow) identified by fMRI is shown together with streamlines (light grey) terminating near CSv. CSv and streamlines are overlaid on the sagittal (left), coronal (middle), and axial (right) sections of T1-weighted image. See S3 Fig for CSv streamlines which were not categorised into major white matter tracts.

CSv streamlines and the major white matter tracts (Fig 2; see S3 Fig for CSv streamlines not categorised into major white matter tracts).

Fig 5 depicts CSv streamlines that were categorised into major white matter tracts (SLF I, the cingulum, and the callosal fibres) in four subjects. We identified CSv streamlines belonging

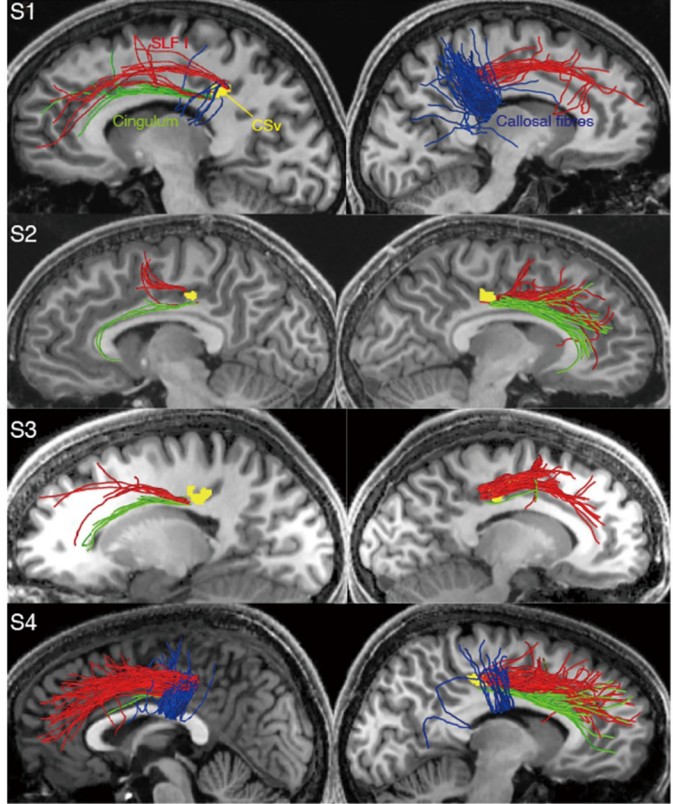

**Fig 5. CSv streamlines which were categorised into major white matter tracts; SLF I (red), the cingulum (green), and the callosal fibres (blue), in the left (left column) and right (right column) hemispheres of four subjects (S1-S4).** CSv (yellow) identified based on fMRI data is shown with the streamlines in each hemisphere. CSv and streamlines are overlaid on a sagittal section medial to CSv and streamlines. Note that CSv is not fully visible in every image as it is located medially to some of the streamlines.

to the dorsal branch of SLF (i.e. SLF I; red in Fig 5) in all but one hemisphere, whereas those belonging to SLF II and SLF III were observed less consistently (14/24 and 11/24 hemispheres, respectively). We also identified CSv streamlines belonging to the cingulum (green in Fig 5) in nearly all hemispheres (22/24). CSv streamlines running into the corpus callosum (i.e. the callosal fibres, blue in Fig 5) were identified in 15/24 hemispheres.

Fig 6 shows the number of CSv streamlines belonging to each of the seven major white matter tracts, expressed as a proportion of the total number of CSv streamlines that were identified, averaged across subjects. Streamlines belonging to SLF I and the cingulum together accounted for approximately 18% of all CSv streamlines. The other tracts examined accounted for a further 6%. The remaining 76% of streamlines may reflect short-range streamlines which do not belong to major white matter tracts; however, anatomical validity of these streamlines is hard to establish because of the limited anatomical knowledge of short-range fibres (see Discussion). For this reason, here we focus on major white matter tracts that are consistently reported in anatomical studies of the white matter.

To test the statistical significance of the presence of CSv streamlines belonging to each tract, we performed one-sample t-tests on the proportion of CSv streamlines belonging to each tract. The proportion of CSv streamlines categorised as SLF I was significantly above zero in the

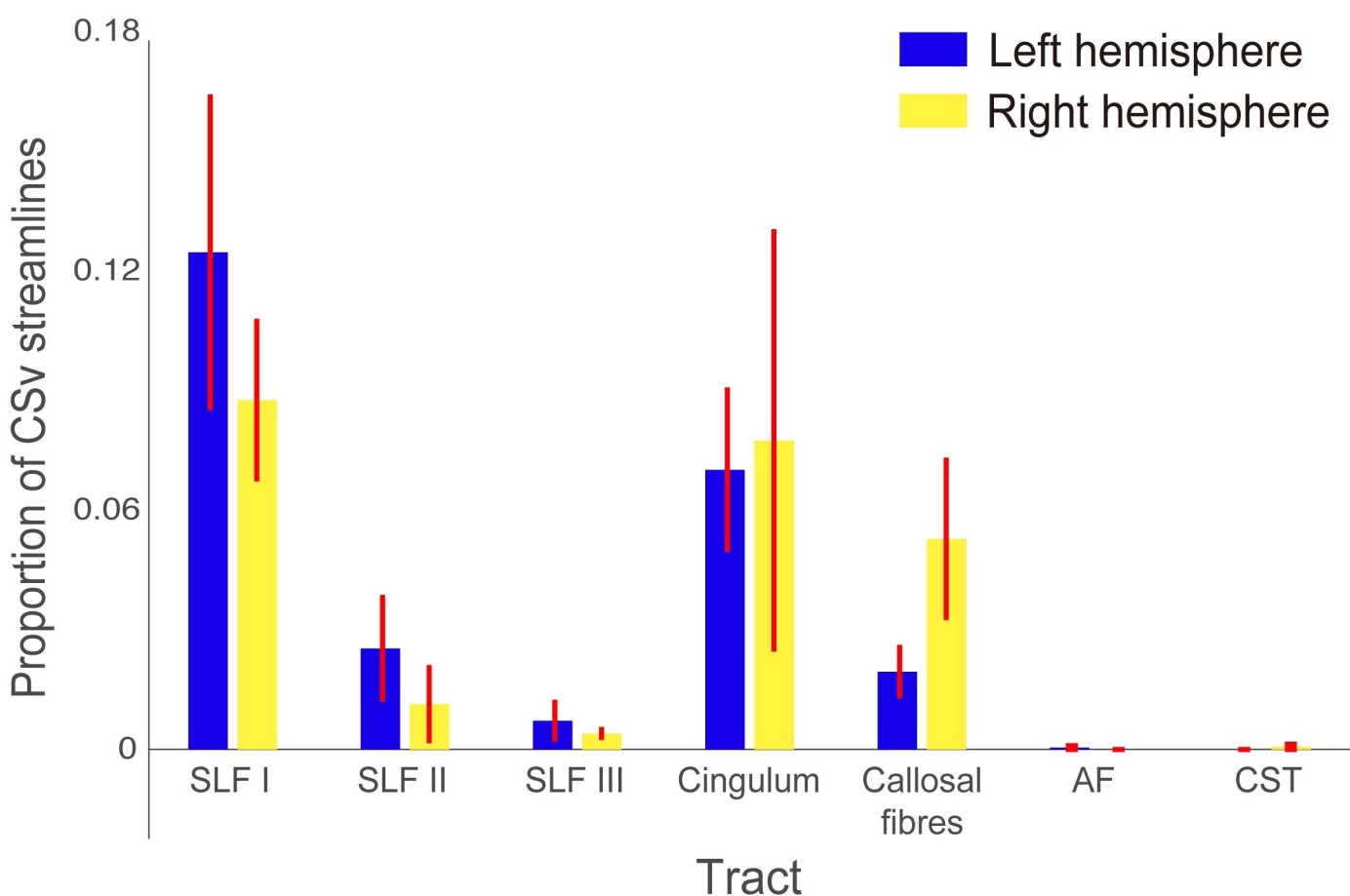

**Fig 6. Summary statistics of the proportion of CSv streamlines belonging to each of the major white matter tracts considered in the left (blue) and right (yellow) hemispheres.** CSv streamlines belonging to SLF I and the cingulum together accounted for approximately 18% of all CSv streamlines. Error bars depict ±1 SEM across subjects. SLF: superior longitudinal fasciculus, AF: arcuate fasciculus, CST: corticospinal tract.

right hemisphere ($t_{11}$ = 4.30, 95% confidence interval [CI] = 0.043–0.133, $P$ = 0.001). We also observed a similar trend for SLF I in the left hemisphere, but the effect did not reach statistical significance when multiple comparisons across tracts were taken into account ($t_{11}$ = 3.15, CI = 0.038–0.211, $P$ = 0.009). The proportion of CSv streamlines belonging to the cingulum was significantly above zero in the left hemisphere ($t_{11}$ = 3.40, CI = 0.025–0.116, $P$ = 0.006), but did not reach statistical significance in the right hemisphere ($t_{11}$ = 1.46, CI = -0.039–0.194, $P$ = 0.172). The proportions of CSv streamlines belonging to the other tracts (SLF II, III, the callosal fibres, AF, and CST) were not significantly larger than zero in either hemisphere ($t_{11}$ = 1.89, CI = -0.004–0.05, $P$ = 0.086 for left SLF II; $t_{11}$ = 1.16, CI = -0.010–0.033, $P$ = 0.271 for right SLF II; $t_{11}$ = 1.37, CI = -0.004–0.019, $P$ = 0.199 for left SLF III; $t_{11}$ = 2.46, CI = 0.0004–0.008, $P$ = 0.025 for right SLF III; $t_{11}$ = 2.90, CI = 0.005–0.034, $P$ = 0.015 for left callosal fibres; $t_{11}$ = 2.59, CI = 0.008–0.098, $P$ = 0.025 for right callosal fibres; $t_{11}$ = 1.00, CI = -5.499–0.002, $P$ = 0.339 for left AF; $t_{11}$ = 1.00, CI = -0.001–0.002, $P$ = 0.339 for right CST). We note that we were unable to perform statistical tests on right AF and left CST, since none of the CSv streamlines were found to belong to these tracts in any of the subjects.

## Discussion

This study aimed to complement the connectivity study of Smith et al. [30] by examining the relationship between CSv streamlines and the known major white matter tracts. Results indicate that notable proportions of CSv streamlines belong to SLF I and the cingulum. Below, we discuss the implications of these findings in relation to neuroanatomy and cortical functions.

### SLF I and CSv

SLF is a white matter tract connecting the frontal and parietal cortices, and can be separated into three branches (SLF I, II, and III) connecting different parts of the frontal and parietal areas [31, 35, 37, 69, 70]. Previous studies demonstrated that the functional significance of these three branches may vary, based on differences in lateralisation, deficits caused by lesions to specific branches, fMRI studies on cortical areas in which the branches terminate, and correlations with behavioural data [35, 36, 71, 72]. Amongst the three branches, SLF I is often discussed in relation to motor functions, as it connects the parietal cortex and supplementary motor area [69], and its lateralisation significantly differs between left- and right-handed individuals [36].

   Results demonstrating that CSv streamlines belong to SLF I, which likely carries motor-related information, have implications for the role of CSv in motor control. Since CSv is activated by visual motion signals compatible with self-motion, it is hypothesised that CSv plays an essential role in integrating sensory and motor information to guide locomotion [17]. Consistent with this notion, a previous fMRI study [15] demonstrated that CSv is activated during leg movements simulating locomotion but not during arm movements. It is plausible that SLF I serves as a pathway conveying information related to motor execution between CSv and the fronto-parietal regions involved in motor processing, such as the supplementary motor area. Such interaction is essential to guiding locomotion.

### Cingulum and CSv

The cingulum is a long association white matter tract which can be divided into at least two components: (i) the dorsal component, which constitutes the white matter adjacent to the cingulate gyrus with the anterior-posterior fibre orientation, and (ii) the ventral component running within the parahippocampal gyrus, retrosplenial cingulate gyrus, and posterior precuneus and eventually reaching the medial temporal cortex [33, 73–75]. CSv streamlines, categorised

as the cingulum, are consistent with the definition of the dorsal subdivision of the cingulum as these streamlines run within the cingulate gyrus. The importance of the cingulum in relation to CSv's function is relatively unclear compared to that of SLF I, since the anterior-dorsal component of the cingulum is often discussed as a pathway subserving the default mode network, attention, memory, and affective functions [76, 77]. It could be speculated, however, that cognitive functions such as memory and decision-making are critical to navigation. In fact, the importance of the cingulum in spatial navigation has been discussed in a lesion study on human patients [78]. A possible interpretation could be that, since the role of CSv in guiding locomotion must ecologically serve an important purpose in navigation, the cingulum provides a channel of signal transmission between CSv and areas involved with spatial memory, which might influence online control and adjustment of locomotory movements during active navigation.

It should also be noted that, according to Schmahmann and Pandya [31], the cingulum terminates in the supplementary motor area in macaques. It is possible that the cingulum is also a motor-related pathway in humans, and has a role in relaying motor-related signals between CSv and the supplementary motor area.

## Relation to macaque studies

Studies on non-human primates, such as rhesus macaques, have complementary strengths as compared with human neuroimaging studies, since invasive methods can be applied to study them, including chemical tracers, which have higher levels of specificity for identifying fibre trajectories from specific injection sites [79, 80]. A comparison of dMRI-based tractography with macaque tracer studies can therefore prove useful not only for validating human findings but also for indicating anatomical features that are shared across species [32, 81].

Schmahmann and Pandya [31] provide an extremely comprehensive atlas of the white matter in rhesus macaque brains, in which an anterograde tracer with radiolabelled isotopes was injected to visualise the trajectories of fibre pathways. In their extensive report, they document fibre pathways near the cingulate gyrus and cingulate sulcus ([31], Fig 12–3, pg. 355). According to their data based on a tracer injection in the prefrontal cortex, the cingulum travels through the white matter regions near the cingulate gyrus. In addition, there are axons from SLF I travelling parallel to the cingulate sulcus and terminating at the cingulate gyrus. Considering the fact that previous fMRI studies on macaques reported the putative macaque homologue of CSv located in the cingulate sulcus [11, 82], those findings may suggest that our results, demonstrating that human CSv is near SLF I and the cingulum, are in line with tracer results by Schmahmann and Pandya [31].

## Conclusions and future directions

In summary, we found that some of the streamlines terminating near CSv belong to major white matter tracts, SLF I and the cingulum, using dMRI-based tractography with anatomical prescriptions. Those findings enable evaluation of how the neuronal circuitry allowing us to monitor and adjust our locomotory movements may be related to white matter tracts known from classical dissection studies, lesion studies, and modern tractography studies. While the possibility cannot be rejected that our results did not statistically support the existence of CSv streamlines belonging to SLF II, III, and the callosal fibres because of the limitations in spatial and angular resolutions of the dataset, it is also possible that the proportions of these tracts terminating near CSv are not as prominent as those of SLF I and the cingulum, and therefore play a limited role in the functions in which CSv is involved.

In this study, we focussed on CSv streamlines that belong to major white matter tracts, rather than the rest including short-range streamlines, since it is difficult to classify streamlines that do not belong to white matter tracts known to exist in anatomical studies. For example, we observed streamlines connecting CSv and the part of the parietal cortex that likely includes the precuneus and the superior parietal lobule (S3B Fig). While these streamlines are of interest as some of the regions involved in optic-flow processing are located in this general area [9, 10, 30], we did not include these streamlines in the main analysis as short-range streamlines are particularly susceptible to partial voluming within the grey matter and to complex fibre crossing [83] and therefore we have limited confidence in the short-range CSv streamlines identified in this study using dMRI data with a standard resolution. Acquisition of dMRI data with novel high-performance gradient systems [84] coupled with the CSv localiser or high-resolution analyses on anatomical data on fibre pathways [85–87] may allow for further analyses of short-range pathways associated with CSv with greater confidence. We believe that an extension of current work, together with improved methods in future studies, will provide a more complete understanding on what type of white matter pathways carry signals between CSv and other cortical regions involved in self-motion processing and motor control. These regions, highlighted by Smith et al. [30], include pVIP (putative intraparietal cortex), hV6, PIC (posterior insular cortex), SMA (supplementary motor area) and the cingulate motor areas. CSv is now seen as a sensorimotor interface in the context of locomotion (see [17] for review); however a full understanding of this system has not yet been reached.

It should also be noted that we took a relatively conservative tractography approach in the present study, applying a filter to the whole-brain streamlines initially generated in order to cull the streamlines that did not account for diffusion signals [41, 44, 57]. It is well established that probabilistic tractography is prone to producing spurious streamlines. Filtering the streamlines has been shown to yield more reliable estimates of structural connectivity [41, 44, 88, 89]. This process removed streamlines including, for example, those connecting CSv and the visual cortex, which were present amongst the streamlines that were initially generated (S2 Fig). This does not necessarily mean that structural connections do not exist between CSv and the visual cortex, however, these streamlines were not sufficiently supported by our dMRI data according to the conservative criteria of the filtering process. As discussed above, it is possible that if streamlines that contribute to genuine connections have been filtered out, similar streamlines may survive the same filtering process if generated from future dMRI data acquired with improved methods.

We used an approach to examine the relationship between CSv and white matter tracts known to exist based on anatomical knowledge. This approach complements the approach used in Smith et al. [30], which focuses on connectivity patterns between grey matter regions without an explicit hypothesis on the existence of white matter tracts [30]. The strength of our approach is that the findings can be used as evidence for validity of tractography results based on anatomical knowledge, and to link a functionally defined area with functional roles of specific white matter tracts known from studies on lesion, lateralisation, and development [33, 35]. Our approach allowed the same set of data as in Smith et al. [30] to map specific portions of CSv streamlines onto the major white matter tracts with known functional roles and therefore more directly implicate CSv in said functions, rather than to speculate the roles of CSv connections based on their cortical endpoints. Our findings that CSv streamlines belong to (i) SLF I, of which involvement in integration of sensory information and motor planning supporting visuospatial attention and complex motor execution is well-documented [90], substantiates CSv's role as an interface between perception and action; while (ii) the cingulum, of which association with cognitive functions such as attention and memory [77], implies CSv's possible involvement in active navigation. Although it is beyond the scope of our study, and

therefore remains speculative, the possibility that CSv may be involved in navigation would be novel if confirmed as it has not been suggested in previous studies (e.g. [17, 20, 30]). We believe that our findings, together with the findings of Smith et al. [30], provide strong evidence that CSv links sensory and motor systems in the context of self-motion and also connote CSv's potential role in navigation.

Better understanding of the tissue properties of SLF I and the cingulum, and their relations with sensory and motor performance or disorders may provide a fuller picture of the role of CSv. Since it has been established that CSv receives not only visual but vestibular input [13, 23, 24], it may also be informative to examine the relations between these white matter tracts and vestibular areas in the medial cortex such as the vestibular pericallosal sulcus [91]. Finally, it would be beneficial for future research to investigate how lesions to SLF I and the cingulum, or individual variation of those tracts correlate with behavioural measurements on self-motion perception, locomotion, and navigation.

## Supporting information

**S1 Fig. Location of CSv in three additional subjects (S2-S4).** CSv (yellow patches highlighted by white circles) overlaid on the sagittal (left), coronal (middle), and axial (right) sections. (PDF)

**S2 Fig.** Comparison between A. all of CSv streamlines generated using ensemble tractography [44] and B. CSv streamlines that remained after the application of LiFE [41, 57] in the left hemispheres of four subjects (S1-S4). Left CSv (yellow) identified by fMRI is shown together with streamlines (light grey) terminating near CSv. CSv and streamlines are overlaid on the sagittal (left), coronal (middle), and axial (right) sections of T1-weighted image. (PDF)

**S3 Fig. CSv streamlines not categorised as part of major white matter tracts.** A. Streamlines not categorised as part of major white matter tracts in the left hemisphere of one representative subject (S1). Conventions are identical to those in Fig 4. B. Short-range streamlines connect CSv with the superior parietal regions. Left CSv (yellow) and the trajectories of streamlines that were not categorised as part of major white matter tracts (orange) are overlaid on a representative coronal section of the T1-weighted volume. Some of the streamlines can be observed connecting CSv and the superior parietal regions, presumably including the precuneus and the superior parietal lobule. (PDF)

## Acknowledgments

We thank Yusuke Sakai for technical assistance.

## Author Contributions

**Conceptualization:** Maiko Uesaki, Michele Furlan, Andrew T. Smith, Hiromasa Takemura.

**Formal analysis:** Maiko Uesaki, Hiromasa Takemura.

**Funding acquisition:** Maiko Uesaki, Hiromasa Takemura.

**Investigation:** Maiko Uesaki, Hiromasa Takemura.

**Methodology:** Hiromasa Takemura.

**Project administration:** Hiromasa Takemura.

**Resources:** Michele Furlan, Andrew T. Smith.

**Software:** Maiko Uesaki, Hiromasa Takemura.

**Supervision:** Andrew T. Smith, Hiromasa Takemura.

**Validation:** Maiko Uesaki, Hiromasa Takemura.

**Visualization:** Maiko Uesaki, Hiromasa Takemura.

**Writing – original draft:** Maiko Uesaki, Hiromasa Takemura.

**Writing – review & editing:** Maiko Uesaki, Andrew T. Smith, Hiromasa Takemura.

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
