## [Decision Letter · Decision Letter 0]

13 Oct 2023

PONE-D-23-21002White matter tracts adjacent to the human cingulate sulcus visual area (CSv)PLOS ONE

Dear Dr. Uesaki,

Thank you for submitting your manuscript to PLOS ONE. After careful consideration, we feel that it has merit but does not fully meet PLOS ONE’s publication criteria as it currently stands. Therefore, we invite you to submit a revised version of the manuscript that addresses the points raised during the review process.

We look forward to receiving your revised manuscript.

Kind regards,

Akitoshi Ogawa, Ph.D.

Academic Editor

PLOS ONE

Journal Requirements:

"This work was supported by Japan Society for the Promotion of Science (JSPS; https://www.jsps.go.jp/) KAKENHI (JP17H04684 and JP21H03789, H.T.), Grant-in-Aid for JSPS Fellows (JP13J05795, M.U.), and Research Grant for Nanyang Technological University Presidential Postdoctoral Fellows (https://www.ntu.edu.sg/research/research-careers/presidential-postdoctoral-fellowship-(ppf))."

"This work was supported by Japan Society for the Promotion of Science (JSPS) KAKENHI (JP17H04684 and JP21H03789, H.T.), Grant-in-Aid for JSPS Fellows (JP13J05795, M.U.), and Research Grant for Nanyang Technological University Presidential Postdoctoral Fellows. We thank Yusuke Sakai for technical assistance."

"This work was supported by Japan Society for the Promotion of Science (JSPS; https://www.jsps.go.jp/) KAKENHI (JP17H04684 and JP21H03789, H.T.), Grant-in-Aid for JSPS Fellows (JP13J05795, M.U.), and Research Grant for Nanyang Technological University Presidential Postdoctoral Fellows (https://www.ntu.edu.sg/research/research-careers/presidential-postdoctoral-fellowship-(ppf)). The funders had no role in study design, data collection and analysis, decision to publish, or preparation of the manuscript."

Reviewers' comments:

Reviewer's Responses to Questions

**Comments to the Author**

1. Is the manuscript technically sound, and do the data support the conclusions?

Reviewer #1: Partly

Reviewer #2: Yes

2. Has the statistical analysis been performed appropriately and rigorously? 

Reviewer #1: Yes

Reviewer #2: No

3. Have the authors made all data underlying the findings in their manuscript fully available?

Reviewer #1: Yes

Reviewer #2: No

4. Is the manuscript presented in an intelligible fashion and written in standard English?

Reviewer #1: Yes

Reviewer #2: Yes

5. Review Comments to the Author

Reviewer #1: In this manuscript, Uesaki and colleagues use diffusion tractography data to ask about the white matter used by streamlines of the human cingulate sulcus visual area (CSv). In a previous work, Smith et al. established the location of the CSv in individual subjects using a functional localizer, then examined the endpoints of streamlines originating there (as well as various functional connectivity properties). Here, using the same dataset, the streamlines are categorized according to known white matter bundles. They mainly show CSv usage of the SLF I and cingulum.

This is a relatively straightforward contribution to the literature; although not a major advance, it is potentially a useful one, and certainly deserves publication. I do think there is a little work to do to really understand how these fibers travel.

The biggest problem is that 75% of the streamlines are unaccounted for. I understand that the authors think that these represent short-range connections, and that their validity is somewhat in question, but more work needs to be done on this problem. I have a few suggestions. First, I think we need to see these streamlines on their own. In other words, essentially reproduce Figure 5 but remove the categorized fibers (the ones belonging to SLF I, cingulum, CC, etc). Then show additional (coronal, horizontal) views to figure out what these uncategorized fibers really are. If they are entirely short-range fibers, some of them are probably quite accurate; after all, cortical regions tend to connect extremely strongly with nearby regions. There may be long-range fibers in there, too, though. It’s possible that some of them ought to be categorized as one of the known bundles, but have been left out by the particular approach used here.

Second, I am looking at Figure 5 and comparing it to the endpoints in Smith et al., and they just really don’t match up. I think that the authors of the current study are somehow throwing away a lot of connectivity identified in Smith et al. Why is that? I don’t see any or many streamlines that could account for the substantial connectivity shown in the earlier paper with V1/V2/V3 or MT, in particular. One solution here would be to look at the endpoints of all the fibers, and then also the fibers categorized as belonging to the known white matter bundles.

Third, it seems like there are known bundles, especially the middle longitudinal fasciculus and the various dorsal-ventral bundles I’m sure this group is quite familiar with, that should be part of this analysis.

These three comments are related. Basically, I think that there are likely connections that need to be accounted for in one way or another (either explain why they are not there, or categorize them more carefully).

Minor comment

-I don’t think Figure 1 is adding much; I would not include it, but would just refer the reader back to Wall and Smith. As the first figure the reader encounters, it just isn’t that relevant.

Reviewer #2: The authors examined the white matter tracks based on diffusion magnetic resonance imaging (dMRI) of 12 healthy young people. The goal was to test the spatial correspondence of tracks connected with the cingulate sulcus visual (CSv) area, an region involved in visual-vestibular processing, with known white matter tracks in the dorsal brain. The authors found that CSv tracks corresponded most strongly with the superior longitudinal fasciculus I (SLF I), the cingulum track, and callosal tracks, but not or less with the SLF II, SLF III.

The study reanalyzes data that was already published (as mentioned in text), but with a different analysis approach. Nevertheless, the findings substantially elaborate this previous work and hence are sufficiently novel and relevant. The analysis is solid. However, some aspects of the manuscript should be elaborated.

Abstract: Please do not cite articles in the abstract, because it may be ambiguous, if no full reference is given. Referring to 'published data' would be sufficient in the abstract. Citing the article in text (introduction, methods, discussion) is necessary and sufficient.

l. 111ff: More details on known tracts is needed. This paragraph needs to be elaborated. For instance, what were the criteria for tract classification? How do the different tract categorization schemes differ or agree? How many tracts do the cited references categorize overall? What were the criteria for selecting SLF I, II, III, cingulum, and callosal fibres and ignoring others? Why were only these five tracts analyzed?

l. 233ff: The waypoint definitions for SLF, cingulum, and callosal fibres must be explained more clearly. Why did the authors define them manually? Is there an automatic procedure and why was it not used? What were the criteria for manual selection (e.g., what landmarks were used)? This needs to be described at sufficient detail that an external researcher could replicate it.

l. 255ff: I am a bit sceptical with testing number of streamlines against zero. In fiber tracking, the number (probability) of streamlines will always be > 0 even in random data. A more careful null hypothesis should be chosen. For instance, testing the number of streamlines between two tracks implicitly takes random streamlines into account. Or the authors could test the number of streamlines for each track against the track with the least number of streamlines.

l. 268: More detailed information on the location and size of CSv should be provided. For instance, Talairach or MNI coordinates of CSv.

l. 371 ff: Is it possible that the cingulum track is related to the recently described vestibular pericallosal sulcus (vPCS) region (dx.doi.org/10.1152/jn.00431.2022)? The authors should discuss whether such a relationship is possible (or alternatively why it is not possible).

Fig. 1: This figure reproduces already published work. It is not essential and paritally redundant. It could be combined with Fig. 2 as a sub-panel. I even recommend to drop this figure completely.

Fig. 8: Although I am a bit sceptical regarding reproducing other people's work, this figure may be acceptable. However, permission must be obtained prior to acceptance. Moreover, all abbreviations must be explained in the figure legend. The relationship to the current findings must be shown more clearly. For instance, it is hard to decipher whether the dotted lines of red dots correspond to SLF I or the cingulum tract. Please, be more specific. Using different colors for the two tracts might help.

6. PLOS authors have the option to publish the peer review history of their article (what does this mean?). If published, this will include your full peer review and any attached files.

Reviewer #1: No

Reviewer #2: No

---

## [Author Response · Author response to Decision Letter 0]

31 Jan 2024

Please see the uploaded document entitled "Response to Reviewers".

---

## [Decision Letter · Decision Letter 1]

1 Mar 2024

White matter tracts adjacent to the human cingulate sulcus visual area (CSv)

PONE-D-23-21002R1

Dear Dr. Uesaki,

We’re pleased to inform you that your manuscript has been judged scientifically suitable for publication and will be formally accepted for publication once it meets all outstanding technical requirements.

Kind regards,

Akitoshi Ogawa, Ph.D.

Academic Editor

PLOS ONE

Reviewers' comments:

Reviewer's Responses to Questions

**Comments to the Author**

1. If the authors have adequately addressed your comments raised in a previous round of review and you feel that this manuscript is now acceptable for publication, you may indicate that here to bypass the “Comments to the Author” section, enter your conflict of interest statement in the “Confidential to Editor” section, and submit your "Accept" recommendation.

Reviewer #1: All comments have been addressed

Reviewer #2: (No Response)

2. Is the manuscript technically sound, and do the data support the conclusions?

Reviewer #1: Yes

Reviewer #2: Yes

3. Has the statistical analysis been performed appropriately and rigorously? 

Reviewer #1: Yes

Reviewer #2: Yes

4. Have the authors made all data underlying the findings in their manuscript fully available?

Reviewer #1: Yes

Reviewer #2: No

5. Is the manuscript presented in an intelligible fashion and written in standard English?

Reviewer #1: Yes

Reviewer #2: Yes

6. Review Comments to the Author

Reviewer #1: (No Response)

Reviewer #2: (No Response)

7. PLOS authors have the option to publish the peer review history of their article (what does this mean?). If published, this will include your full peer review and any attached files.

Reviewer #1: No

Reviewer #2: No

---

## [Editor Report · Acceptance letter]

23 Mar 2024

PONE-D-23-21002R1 

PLOS ONE

Dear Dr. Uesaki, 

I'm pleased to inform you that your manuscript has been deemed suitable for publication in PLOS ONE. Congratulations! Your manuscript is now being handed over to our production team.

Kind regards, 

on behalf of

Dr. Akitoshi Ogawa 

Academic Editor

PLOS ONE